# Evaluation of Matrix-Assisted Laser Desorption Ionization–Time of Flight Mass Spectrometry for Molecular Typing of *Acinetobacter baumannii* in Comparison with Orthogonal Methods

Eloise J. Busby,[a] Ronan M. Doyle,[b,c] Clara Leboreiro Babe,[d] Kathryn A. Harris,[e] Damien Mack,[d,f] Gema Méndez-Cervantes,[g] Denise M. O'Sullivan,[a] Vicky Pang,[f] Zahra Sadouki,[d] Priya Solanki,[d] Jim F. Huggett,[a,h] Timothy D. McHugh,[d] Emmanuel Q. Wey[d,f]

[a]National Measurement Laboratory, LGC, Teddington, Middlesex, United Kingdom

[b]Department of Microbiology, Virology and Infection Control, Great Ormond Street Hospital for Children NHS Foundation Trust, London, United Kingdom

[c]Clinical Research Department, London School of Hygiene & Tropical Medicine, London, United Kingdom

[d]Centre for Clinical Microbiology, Royal Free Campus, Division of Infection and Immunity, Faculty of Medical Sciences, University College London, London, United Kingdom

[e]Virology Department, ESEL Pathology Partnership, Royal London Hospital, Barts Health NHS Trust, London, United Kingdom

[f]Royal Free Hospital NHS Foundation Trust, London, United Kingdom

[g]Clover Bioanalytical Software, Granada, Spain

[h]School of Biosciences & Medicine, Faculty of Health & Medical Science, University of Surrey, Guildford, United Kingdom

Clara Leboreiro Babe and Priya Solanki contributed equally to this work.

**ABSTRACT** Colonization and subsequent health care-associated infection (HCAI) with *Acinetobacter baumannii* are a concern for vulnerable patient groups within the hospital setting. Outbreaks involving multidrug-resistant strains are associated with increased patient morbidity and mortality and poorer overall outcomes. Reliable molecular typing methods can help to trace transmission routes and manage outbreaks. In addition to methods deployed by reference laboratories, matrix-assisted laser desorption ionization–time of flight mass spectrometry (MALDI-TOF MS) may assist by making initial in-house judgments on strain relatedness. However, limited studies on method reproducibility exist for this application. We applied MALDI-TOF MS typing to *A. baumannii* isolates associated with a nosocomial outbreak and evaluated different methods for data analysis. In addition, we compared MALDI-TOF MS with whole-genome sequencing (WGS) and Fourier transform infrared spectroscopy (FTIR) as orthogonal methods to further explore their resolution for bacterial strain typing. A related subgroup of isolates consistently clustered separately from the main outbreak group by all investigated methods. This finding, combined with epidemiological data from the outbreak, indicates that these methods identified a separate transmission event unrelated to the main outbreak. However, the MALDI-TOF MS upstream approach introduced measurement variability impacting method reproducibility and limiting its reliability as a standalone typing method. Availability of in-house typing methods with well-characterized sources of measurement uncertainty could assist with rapid and dependable confirmation (or denial) of suspected transmission events. This work highlights some of the steps to be improved before such tools can be fully integrated into routine diagnostic service workflows for strain typing.

**IMPORTANCE** Managing the transmission of antimicrobial resistance necessitates reliable methods for tracking outbreaks. We compared the performance of MALDI-TOF MS with orthogonal approaches for strain typing, including WGS and FTIR, for *Acinetobacter baumannii* isolates correlated with a health care-associated infection (HCAI) event. Combined with epidemiological data, all methods investigated identified a group of isolates that

Address correspondence to Emmanuel Q. Wey, Emmanuel.wey@nhs.net.

The authors declare a conflict of interest. Emmanuel Wey is on the advisory board for Clover Bioanalytical Software.

were temporally and spatially linked to the outbreak, yet potentially attributed to a separate transmission event. This may have implications for guiding infection control strategies during an outbreak. However, the technical reproducibility of MALDI-TOF MS needs to be improved for it to be employed as a standalone typing method, as different stages of the experimental workflow introduced bias influencing interpretation of biomarker peak data. Availability of in-house methods for strain typing of bacteria could improve infection control practices following increased reports of outbreaks of antimicrobial-resistant organisms during the COVID-19 pandemic, related to sessional usage of personal protective equipment (PPE).

**KEYWORDS** *Acinetobacter*, MALDI-TOF MS, molecular subtyping

Carbapenem-resistant organisms (CRO), including *Acinetobacter baumannii*, are a significant threat to patients within intensive care units (ICU) and on oncology wards and those who are immunocompromised. *A. baumannii* forms biofilms, colonizes the respiratory tract, and poses a significant transmission risk within the hospital setting (1). *A. baumannii* is responsible for an increasing number of difficult-to-treat respiratory, soft tissue, and bloodstream infections. Risk factors for *A. baumannii* colonization and subsequent bacteremia, particularly with multidrug-resistant (MDR) organisms, include length of hospital stay, ICU admission, and having an intravenous catheter or ventriculoperitoneal (VP) shunt (2, 3). Minimizing nosocomial transmission of *A. baumannii* among vulnerable patient cohorts is critical, and reliable methods for molecular strain typing of *A. baumannii* isolates can assist with determining transmission routes and tracing outbreaks (4).

Methods favored by many bacterial reference laboratories for molecular typing of *A. baumannii* include pulsed-field gel electrophoresis (PFGE), multiple-locus variable-number tandem-repeat (VNTR) analysis (MLVA), and multilocus sequence typing (MLST) (5). However, these methods are subjective in their interpretation, labor-intensive, and costly with lengthy turnaround times for hospitals to receive results. Currently, two MLST schemes exist for typing *A. baumannii* (6, 7) with only three out of seven housekeeping gene targets in common (8), potentially making harmonization of *A. baumannii* typing difficult between schemes. Matrix-assisted laser desorption ionization–time of flight mass spectrometry (MALDI-TOF MS) is an established and widely used method for identifying bacterial species and has been reported as a promising technique for strain typing bacteria (9–11). MALDI-TOF MS has been successfully implemented to resolve nosocomial and foodborne outbreak-associated bacterial strains (12–16), including for *A. baumannii* (17). However, there is limited evidence supporting the robustness of MALDI-TOF MS in this context, and conflicting evidence exists suggesting that MALDI-TOF MS is unreliable and unsuitable for strain typing *A. baumannii* (18–20). Development of in-house capability for strain typing using modified MALDI-TOF MS protocols could improve outbreak surveillance, an attractive option as many clinical microbiology services already possess the necessary instrumentation.

Whole-genome sequencing (WGS) is also gaining traction for bacterial strain typing as an alternative to contemporary approaches (21). WGS is increasingly being applied by reference laboratories for strain typing *A. baumannii* (22–26), as well as becoming more accessible to routine clinical laboratories for in-house applications (27, 28). WGS provides a broader analysis, offering information on drug resistance and a full spectrum of genes, therefore facilitating better resolution between strains. Fourier transform infrared spectroscopy (FTIR) has also demonstrated potential as an emerging application for strain typing bacteria in the clinical laboratory setting. Recent studies have highlighted a potential role for FTIR to recognize clonal relationships between isolates of various species, owing to high discriminatory power offered by the technique (15, 29–31). These emerging molecular approaches could be applied as orthogonal techniques alongside MALDI-TOF MS for supporting timely, in-house identification of nosocomial outbreaks of *A. baumannii*.

Our study explored the technical robustness of MALDI-TOF MS for bacterial strain typing. We also compared outputs from different MALDI-TOF MS data analysis methods

with WGS and FTIR to evaluate typing resolution on a cohort of samples associated with a health care-associated infection (HCAI) transmission event. Different methods measuring various biomarkers including genomic loci, ribosomal proteins, lipids, and surface glycoproteins will offer different levels of resolution for strain typing challenging organisms such as *A. baumannii*. Demonstrating the reliability and evaluating the applicability of MALDI-TOF MS strain typing through comparison with orthogonal methods could support its incorporation into routine typing practices. This may shorten turnaround times to confirm or rule out nosocomial outbreaks, streamlining infection control and treatment options.

## RESULTS

**MALDI-TOF MS strain typing of *A. baumannii*.** The Bruker MALDI Biotyper protocol was applied to 18 OXA-23 clone 1 HCAI outbreak-associated *A. baumannii* isolates. Although a single practical approach was followed, data were analyzed using multiple methods: Bruker FlexAnalysis to analyze spectra and subsequently two separate bioinformatics software programs for analysis of exported peak data. To evaluate MALDI-TOF MS as a robust typing method, 27 spectra were obtained for each isolate across a total of 3 days. The Bruker FlexAnalysis method involved visual identification of peaks that could represent strain-specific biomarkers. Four observed peaks that satisfied the inclusion criteria for strain typing were identified (see Table S2a in the supplemental material), and the isolates were grouped into a total of 8 classes (labeled biotypes A to H) representing different "MALDI biotypes" (Table S2b). Two of the isolates (MBT16-015 [biotype G] and MBT16-030 [biotype H]) were unique biotypes, as their MALDI-TOF MS profiles did not match those of any of the other isolates following this analysis approach.

Summarized spectra for each isolate were subsequently analyzed using BioNumerics software. Two main clusters were identified using this method, designated group I and group II. Of note, the isolates that clustered into group II using the bioinformatics analysis method also fell within the same MALDI biotype group B designated in Table S2b (isolates MBT16-005, MBT16-011, and MBT16-039). This finding was confirmed when the same spectra were clustered using additional data analysis software (Fig. S1).

**Peak intensity as a metric for discrimination of peak classes using MALDI-TOF MS typing.** We examined peak height (intensity) to ascertain whether improved reproducibility of spectra enables better-defined MALDI-TOF MS strain typing resolution. Peak intensity is reported to be a useful indicator of ionization efficiency in MALDI-TOF MS (32) and could be indicative of unique intraexperimental matrix-analyte interactions, laser energy, crystal morphology, and detector performance (33). For peaks that were shared among the 18 HCAI *A. baumannii* isolates (*m/z* 5178 and 5751 determined using BioNumerics), the isolates designated group II exhibited the lowest relative standard deviation (RSD) compared to the other isolates (<0.30 RSD). However, when this approach was applied to a peak at *m/z* 3723, identified by BioNumerics as unique to these three isolates, the mean RSD was up to 10-fold higher, indicating a decrease in reproducibility between peak height values. Regardless of peak class, there was an overall trend of increasing variability in peak height at progressive stages of the experiment protocol, with technical replicates exhibiting the lowest variability and between-day replicates exhibiting the highest (Fig. 1).

**Comparison of methods for strain typing *A. baumannii*. (i) HCAI event *A. baumannii* isolates.** The 8 user-defined MALDI biotypes (Table S2b) and 2 bioinformatic MALDI-TOF MS groups were compared with WGS and FTIR typing (Fig. 2). There was limited correlation between the MALDI biotypes and the other methods except for the biotype B isolates (MBT16-005, MBT16-011, and MBT16-039), which consistently clustered together for all four methods. The bioinformatic MALDI-TOF MS analysis and WGS clustered the isolates into two main clades, whereas FTIR identified three clusters for the 18 outbreak isolates (331, 328, and 323). The FlexAnalysis method appeared to overestimate the isolate diversity compared to the other methods, with 8 biotypes being identified based on visual inspection of spectra (Fig. 2a, Key i). The group I isolates were shown by MALDI-TOF MS bioinformatics analysis methods and WGS to be

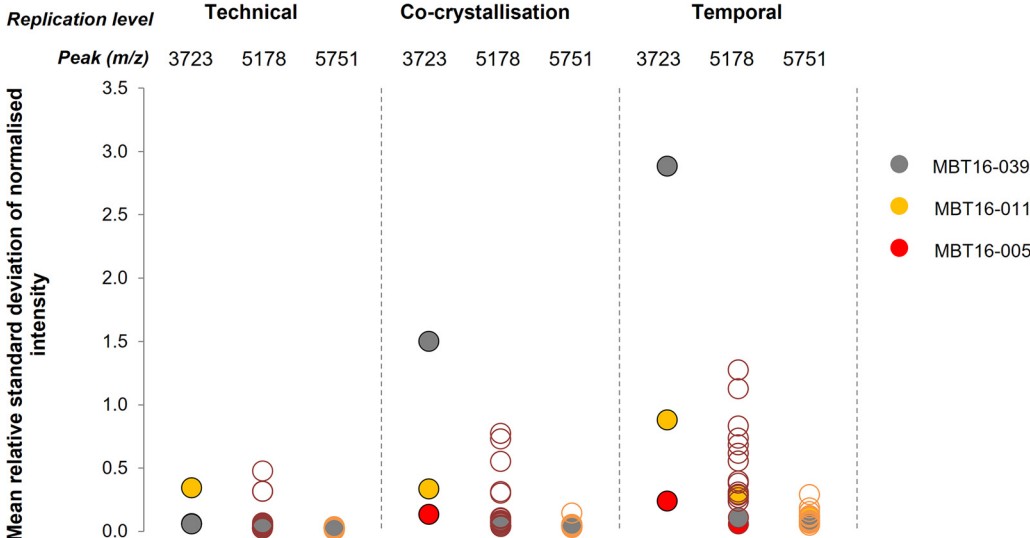

**FIG 1** Variability in peak height at different stages of the MALDI-TOF MS typing protocol defined as (i) technical (where spectra were acquired from individual spots of protein extract in triplicate and treated as distinct data sets), (ii) cocrystallization (separate 1-$\mu$L spots of protein extract deposited on a MALDI target plate in triplicate and overlaid with HCCA matrix), and (iii) temporal (the day-to-day variability between experiments, incorporating subculture of isolates onto fresh agar). The group II/MALDI biotype B isolates are represented by the filled circles: red, MBT16-005; yellow, MBT16-011; and gray, MBT16-039. Unfilled circles represent the other 15 isolates. *m/z* 5178 and 5751 represent peak classes common to all isolates; *m/z* 3723 represents a peak observed only for the group II/MALDI biotype B isolates following analysis in BioNumerics.

highly related to one another. The FTIR approach assigned 11 out of 15 MALDI bioinformatics (Fig. 2a) and 10 out of 14 WGS (Fig. 2b) group I isolates to cluster 328. The isolates that were assigned to group II by MALDI-TOF MS and WGS were identified as cluster 331 by FTIR, along with isolate MBT16-003.

**(ii) Comparison with temporally and geographically linked *A. baumannii* isolates.** Figure 3 shows that the HCAI outbreak isolates clustered together with a high degree of genetic relatedness compared to the nonoutbreak isolates on other branches following WGS analysis. Some of the reference isolates (Abau-Iso2-11, -19, -20, -23, -26, and -27) clustered with the outbreak cohort, whereas the remaining Abau-Iso2 isolates were genetically divergent from the outbreak group. Two isolates, Abau-Iso2-15 and -16, were temporally distinct from the others as they were collected in 2011 and 2012, respectively, and clustered separately from the other organisms. Single nucleotide variant (SNV) analysis (see Files S1 and S2 in the supplemental material) revealed that the three group II isolates (MBT16-005, -011, and -039) were closely related, although containing a greater number of SNVs than the other isolates. Unexpectedly, isolate Abau-Iso2-11 contained a maximum of 2 SNVs compared to these three isolates despite being a nongeographically and nontemporally associated isolate selected for comparison in this study.

The 18 HCAI outbreak isolates were assigned to three clusters following FTIR analysis, designated 328, 331, and 323 (Fig. 2). Compared to the 13 reference isolates, the HCAI outbreak cohort clustered together as observed following WGS analysis, and these isolates were distinct from the other organisms (Fig. 4). This largely reflects the results presented in Fig. 3, with some discrepancies: Abau-Iso2-11, which clustered with the main group of isolates following WGS with minimal SNV differences compared to the three group II isolates, did not cluster within the same clade as these isolates following FTIR. Additionally, each of the five technical replicates of the reference isolates clusters succinctly with one another in Fig. 4, but this is not the case for the quintuplicates for the 18 HCAI outbreak-associated isolates. This may be reflective of a high degree of similarity in the absorption spectra between these isolates, making them indistinguishable from one another by FTIR analysis.

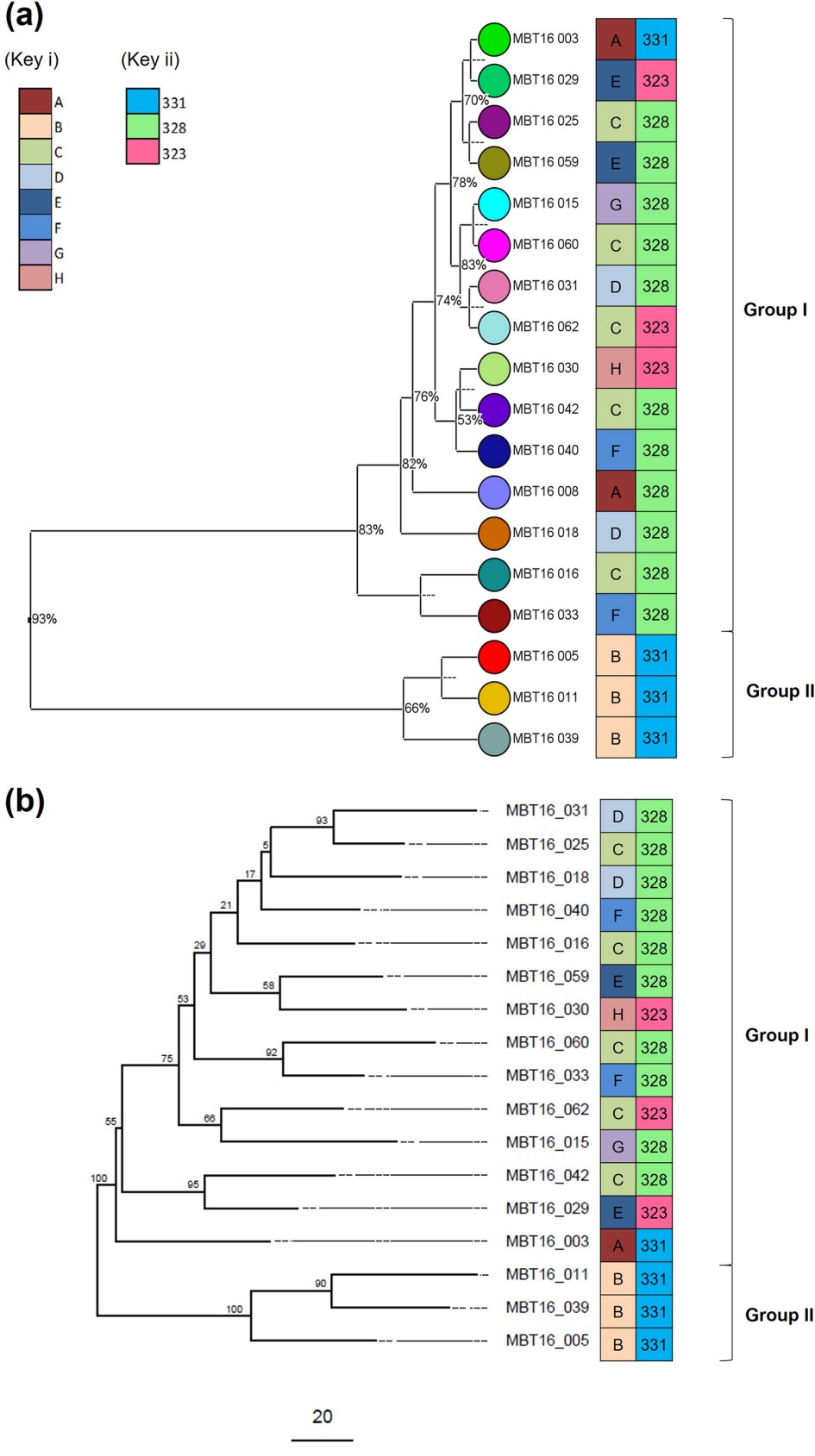

**FIG 2** Comparison of 18 HCAI outbreak-associated *A. baumannii* isolates. (a) UPGMA hierarchical clustering of MALDI-TOF MS spectra generated using BioNumerics software, compared with MALDI biotypes A to H

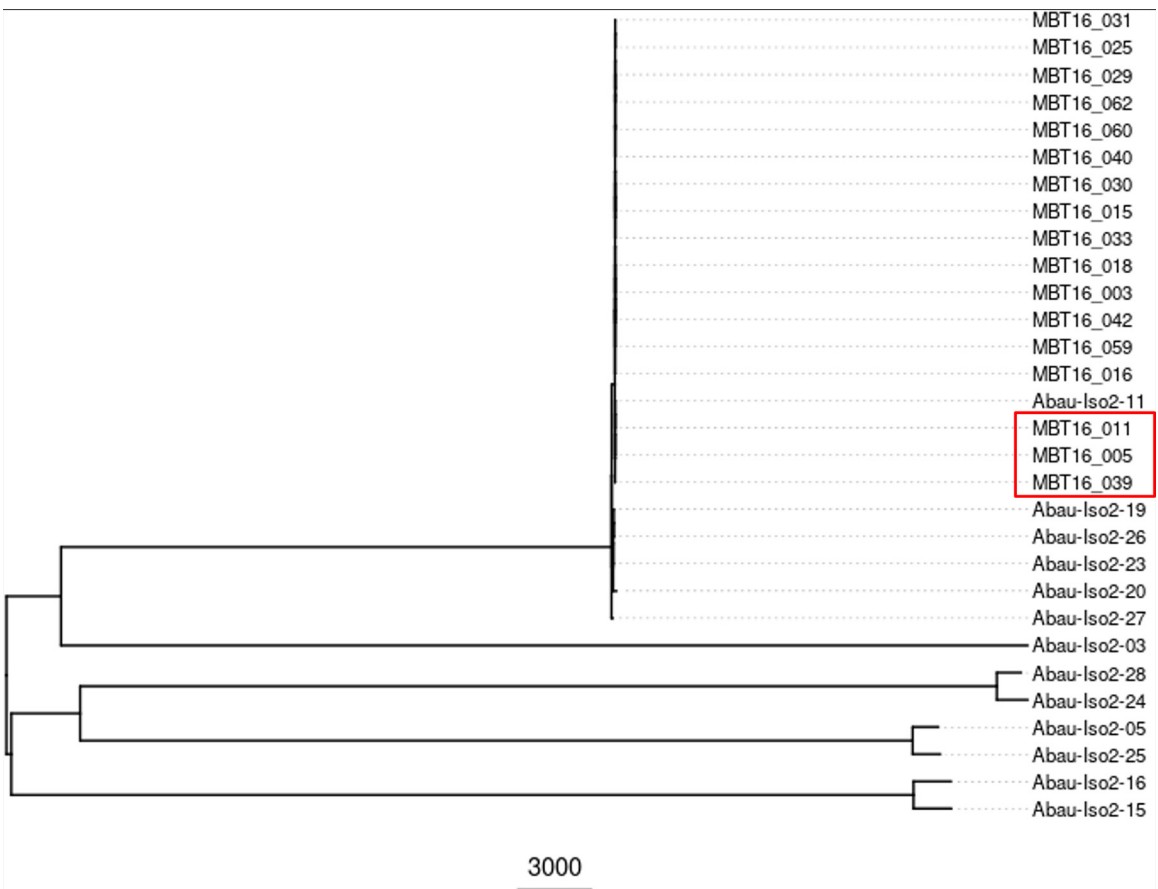

**FIG 3** WGS SNV analysis of 17 of the HCAI outbreak (MBT16) isolates and the 13 additional reference (Abau-Iso2) *A. baumannii* isolates. Note that one of the HCAI outbreak isolates (MBT16-008) failed to grow and therefore could not be analyzed by WGS. The positions of the three group II isolates (Fig. 2) in the tree are indicated by the red box.

## DISCUSSION

**Data analysis method and peak matrix algorithm play a key role in MALDI-TOF MS performance for strain typing *A. baumannii*.** An initial aim of our study was to determine whether MALDI-TOF MS possessed sufficient reproducibility and resolution to identify biomarker peaks that could be used to strain type clinical isolates of *A. baumannii*. The Bruker MALDI Biotyper protocol was applied to a cohort of closely related isolates, and strain typing was performed using different analysis methods. The first method, the Bruker FlexAnalysis method, involved visual inspection of peak classes. This approach yielded eight MALDI biotypes for the isolates that were assigned based on the presence or absence of four peak classes (see Table S2 in the supplemental material). This approach has been described previously for strain typing with various levels of success (34, 35). Overall, there was poor correlation with the other methods, except for three isolates which formed a distinct cluster for all methods tested in this study (Fig. 2). The FlexAnalysis method was based on nonnormalized spectral data, which may result in subjective classification of peaks owing to variable baseline signals

**FIG 2** Legend (Continued)

(Key i) and IR Biotyper cluster (Key ii). Each isolate in the MALDI TOF-MS UPGMA dendrogram is assigned a unique colored spot that is distinct from the color assigned to the other two techniques. Analysis of MALDI TOF-MS spectra was repeated using Clover MS data analysis software, giving results comparable to the BioNumerics output (see Fig. S1 in the supplemental material). (b) WGS SNV analysis compared with MALDI biotypes A to H (Key i) and FTIR cluster (Key ii). The scale bar at bottom refers to the number of single nucleotide differences, and the values on the branches refer to the confidence score for that branch. Note that one of the HCAI outbreak isolates (MBT16-008) failed to grow and therefore could not be analyzed by WGS.

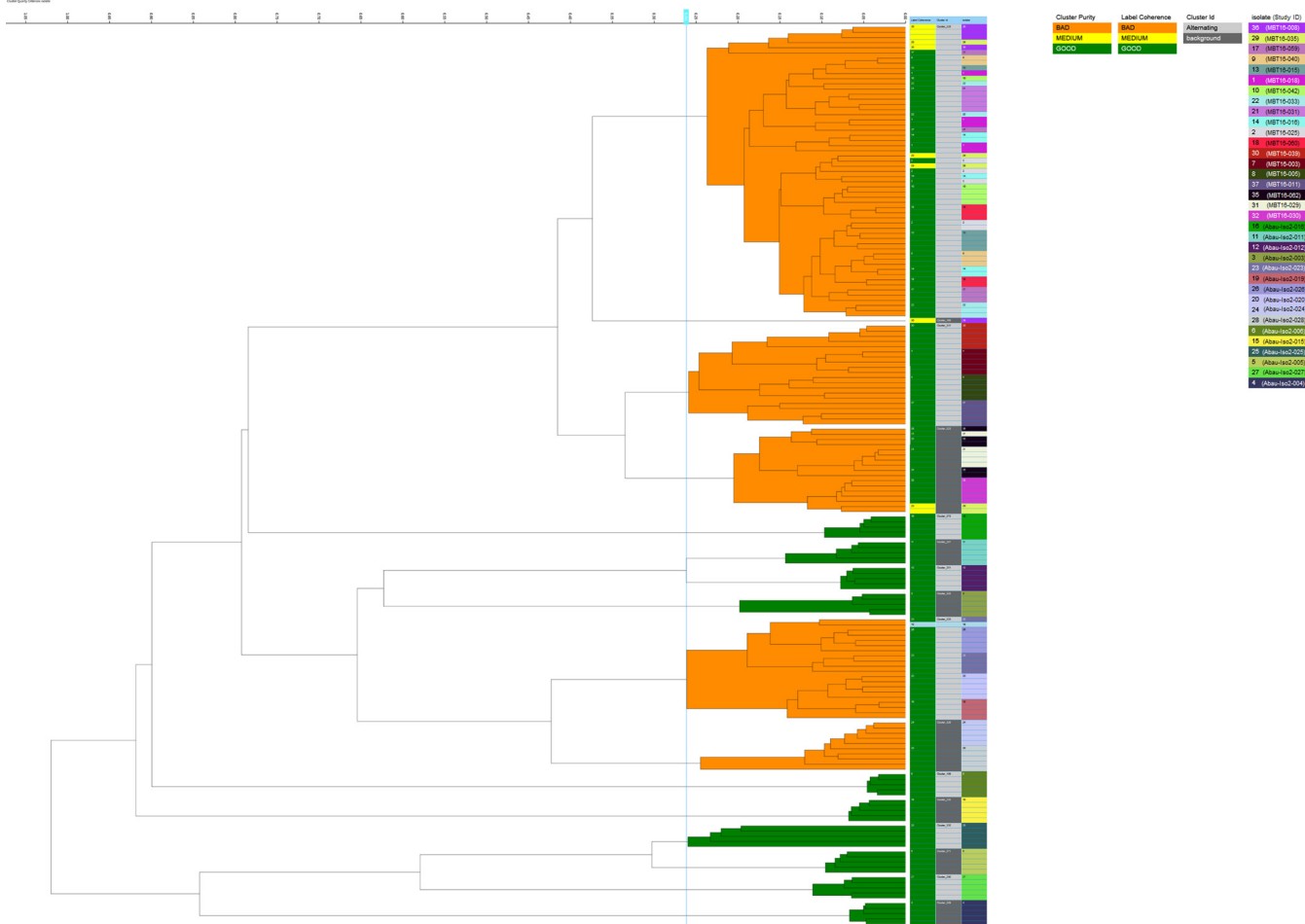

**FIG 4** FTIR analysis comparing the 18 HCAI outbreak isolates and the 13 additional reference *A. baumannii* isolates. Each isolate is represented by a unique color, indicated in the key. Five technical replicates for each isolate are presented in the dendrogram.

between spectra. In addition, resolution may be limited where typing is based on only four out of approximately nine observable peaks per spectrum, which has been reported previously for *A. baumannii* (19, 36).

The MALDI-TOF MS spectra were also analyzed using BioNumerics bioinformatic software, which offered more objectivity as the software can access peak data for the entire spectrum rather than the manual, subjective selection of four classification peaks for the FlexAnalysis method. The software is also able to apply peak height normalization algorithms, which may enable a more robust and standardized approach to typing through removal of operator bias and as a result improve interpretation of spectra. The bioinformatics analysis workflow was repeated for the MALDI-TOF MS spectra using additional software (Clover MS data analysis software), which demonstrated good agreement with the BioNumerics method in terms of isolate clusters (Fig. S1). Several peaks identified using the MALDI-TOF MS bioinformatics workflows were attributed to published ribosomal proteins for *A. baumannii* (Table S3). This suggests that the peak finding algorithms of these methods are fit-for-purpose for finding reference peaks, increasing confidence that the method can be applied to identify potential peaks for strain typing. Further work to explore the capability of additional software for data analysis could be of benefit for future studies on spectral typing methods and for using MALDI-TOF MS to find peaks attributed to drug resistance.

Following bioinformatic analysis of spectra, we evaluated how different stages of the experimental protocol could influence typing of *A. baumannii* due to the respective impact upon peak identification. The quality of MALDI-TOF MS spectra can be influenced

by sample preparation steps, matrix choice, instrumental performance, and analysis methods, among other factors (37, 38). Our chosen metric, peak height (intensity), has been correlated with ionization efficiency (32), which can be influenced by variability in the experimental protocol. Figure 1 shows that peak height became increasingly variable between cocrystallization and day-to-day measurement. The height of peaks common to all *A. baumannii* isolates (e.g., 5751 *m/z*) appeared to be reproducible across experiments for the three isolates that clustered away from the main group (MBT16-005, -011, and -039). However, a peak chosen to represent a biomarker specific to these three isolates (3723 *m/z*) exhibited higher variability, with some individual spectra failing to be called by the software. This suggests that variability introduced during sample preparation could directly influence discrimination of isolates by certain peak classes. Standardizing MALDI-TOF MS workflows could permit better typing resolution (35, 39, 40). Future work incorporating a cohort of diverse strains, along with additional environmental parameters, could provide an opportunity to further evaluate the reproducibility of MALDI-TOF MS typing for a larger number of peak classes.

Our results suggest that using the FlexAnalysis method to analyze MALDI-TOF MS spectra is not appropriate for strain typing *A. baumannii*. The method appeared to overestimate the diversity in a group of closely related isolates and lacked robustness, as it was based on subjective interpretation of a limited number of peaks. Bioinformatic analysis using BioNumerics and Clover MS data analysis software permitted a more objective interpretation of peak classes and enabled the detection of reference proteins in the isolates. In addition, BioNumerics peak matrix analysis enabled metrics for determining robustness of peak detection to be evaluated, which may permit better characterization of reproducibility between analyses. Both the BioNumerics and Clover MS data analysis software platforms were of equivalent cost and subjective ease of use with their respective strengths and weaknesses. Bioinformatic analysis approaches of this kind may be of benefit when handling MALDI-TOF MS strain typing data in the future (35, 40).

**Potential identification of a nosocomial transmission event using four independent methods.** We compared the MALDI-TOF MS FlexAnalysis and bioinformatics *A. baumannii* strain typing results with WGS and FTIR (Fig. 2). The WGS and MALDI-TOF MS bioinformatics approaches clustered the isolates into two groups, with the group I isolates appearing to be closely related, in line with reference laboratory interpretation (Table S1). Any observed differences, such as MALDI-TOF MS peak classes assigned by bioinformatics software or SNV differences by WGS, may represent the typical diversity between isolates of the same strain. The outbreak in question in this study was associated with a single ward (ward A), multiple beds of which were occupied by the patients during this time period (Fig. S2a). However, patients had also spent time on other wards of various specialties within the hospital, including intensive care units. There were temporal overlaps in which patients stayed on ward A, presenting possible opportunities for transmission to occur. High levels of patient migration within the hospital setting necessitate timely and adequate infection control strategies, along with accurate typing methods that can help to quickly confirm or rule out a potential outbreak. Three isolates clustered separately from the main group using all methods (respectively designated group II/cluster 331/MALDI biotype B by MALDI-TOF MS bioinformatics and WGS/FTIR/MALDI-TOF MS FlexAnalysis method). These isolates were also identified as belonging to international clone II lineage OXA-23 clone 1 following reference typing. However, according to multiple methods applied in our study diversity could exist between groups I and II, indicating that they are not identical and possibly not from the same transmission route. Isolates MBT16-005 and MBT16-039 were obtained from the same patient (patient identifier [ID] 2, Table S1). Epidemiological information confirmed isolate MBT16-011 was obtained from another individual (patient ID 4, Table S1) who stayed on ward A in the same male 4-bed bay during this period. The patients occupied opposing beds during the same time frame (Fig. S2b). Multidrug-resistant (MDR) *A. baumannii* was identified first in patient 4, followed 13 days later by patient 2. Unexpectedly, following WGS analysis isolate Abau-Iso2-11 was found to contain a maximum of two SNVs compared to these three isolates despite being randomly selected for this study. The patient was never associated in time

or place with the outbreak; however, there is evidence that this isolate is also a member of the circulating international clone II lineage.

Using four different methods available to laboratories undertaking routine microbiological diagnostics, we have been able to distinguish two temporally and geographically linked transmission clusters within the HCAI cohort, in addition to identifying a related isolate unrelated to the outbreak in question. This has altered the understanding of the outbreak and could have implications for patient management and infection control. Application of these methods as prescreening tools within routine microbiology laboratories could inform clinical decisions while awaiting reference laboratory typing results. This holds particular relevance since the onset of the COVID-19 pandemic, where shortages of personal protective equipment (PPE) and subsequent sessional usage, along with decreased laboratory processing capacity and low health-care-worker (HCW)-to-patient ratios, may have contributed to increased nosocomial transmission of drug=resistant organisms (41, 42).

**Conclusion.** Our study compared MALDI-TOF MS with orthogonal molecular methods for bacterial strain typing in clinical diagnostic laboratories, which may wish to acquire in-house capabilities for prescreening of possible transmission events. The identification of a distinct clonal group that was temporally and geographically linked to a larger HCAI outbreak using MALDI-TOF MS, WGS, and FTIR in our study suggests that a combination of molecular tests, along with bioinformatic analyses and epidemiological data, could help to assign *A. baumannii* strain types more reliably. This could be applied to screen samples and confirm or rule out suspected transmission events prior to sending isolates for reference laboratory typing. Application of MALDI-TOF MS for strain typing should be considered only when paired with bioinformatic data analysis tools. We have demonstrated that FlexAnalysis, the default software for MALDI-TOF MS analysis, overestimated the diversity between related isolates, which could bias interpretation of whether a transmission event has occurred or not. Furthermore, we have empirically demonstrated that the MALDI-TOF MS experimental protocol introduces variability at different stages, which may impact resolution of the technique when awarding biomarker status to a particular peak class. This variability becomes easier to characterize using automated data analysis processes. Further work utilizing a larger number of diverse organisms would help to uncover the potential for emerging strain typing applications, such as FTIR, to reduce the time taken to make clinical decisions while awaiting reference laboratory results, providing an economic solution and overall benefit to patient care.

## MATERIALS AND METHODS

**Drug-resistant clinical isolates.** Eighteen isolates of multidrug-resistant (MDR) *Acinetobacter baumannii* associated with a clinically defined outbreak in accordance with UK Health Security Agency (UKHSA) guidance (43) at the Royal Free London NHS Foundation Trust (Royal Free London) were collected from 15 patients between 2014 and 2015 as part of the routine microbiological diagnostic service. The isolates are listed in Table S1 in the supplemental material along with year collected, PFGE and/or VNTR profile, and patient ID in this study. The outbreak was associated with a single ward (referred to in the present study as ward A); however, patients had collectively migrated between 15 wards over the 2-year period, including an intensive/critical care unit (ICU/CCU) and an outpatient department (OPD). Length of stay on ward A varied between 1 and 184 days. The isolates were confirmed as *A. baumannii* species using the MALDI-TOF Microflex LT (Bruker UK) following the manufacturer's protocol (44). Isolates were assigned a unique study identifier to remove patient information (e.g., MBT16-001).

Thirteen additional randomly selected MDR *A. baumannii* clinical isolates collected at the Royal Free London, which were unrelated to the health care-associated infection (HCAI) outbreak, were included in the analyses to add context to the relatedness of the HCAI isolates (Table S1). These nonoutbreak isolates (termed "reference" isolates in our study) were collected between 2011 and 2017, extending temporally before and after the time period of the HCAI outbreak but not part of the outbreak patient cohort, and assigned the prefix "Abau-iso2" in our study. Three of these isolates (Abau-iso2-11, Abau-iso2-19, and Abau-iso2-20) were collected in 2017 as part of the INHALE study (45).

Antimicrobial susceptibility testing (AST) for all isolates was implemented as part of the routine microbiological diagnostic service according to European Committee on Antimicrobial Susceptibility Testing (EUCAST) breakpoint guidelines (46). Isolates were stored on Cryobank beads (ThermoFisher Scientific Inc.) in glycerol at −80°C.

**Reference laboratory strain typing of isolates.** As part of the routine microbiological service, isolates associated with the HCAI event were sent on nutrient agar slopes for reference laboratory characterization. Pulsed-field gel electrophoresis (PFGE) and variable-number tandem-repeat (VNTR) profiling were performed at four loci (1, 10, 845, and 3468) (5, 47). All 18 isolates were classified as belonging to

international clone II lineage OXA-23 clone 1. PFGE data and VNTR profiles are included in Table S1. Isolate MBT16-062 was not sent to the reference laboratory but was included in this study for prospective analysis. The authors were blind to the reference laboratory typing results prior to commencing experimental work.

**MALDI-TOF MS strain typing protocol.** Strain typing was performed using a MALDI-TOF Microflex (Bruker UK) according to the Bruker MALDI Biotyper protocol described previously (34). The 2014–2015 HCAI outbreak isolates ($n = 18$) were recovered from −80°C onto prepoured Columbia blood agar containing 5% horse blood (ThermoFisher Scientific) and incubated aerobically at 37°C for 24 h. Measurements were obtained using Bruker flexControl software (version 3.4). Triplicate spots of formic acid-extracted protein solution were overlaid with 1 $\mu$L fresh $\alpha$-cyano-4-hydroxycinnamic acid (HCCA) matrix (Bruker UK), allowed to air dry, and measured in triplicate over three separate days to obtain 27 spectra per isolate using freshly cultured isolates each day. Spectra were recorded in positive linear mode within the range of 2 and 20 kDa. External calibration of each MALDI-TOF MS typing experiment was through measurement of bacterial test standard (BTS) solution (Bruker UK).

**Whole-genome sequencing.** DNA was extracted from 30 of the 31 isolates with an UltraClean microbial DNA isolation kit (Mo-Bio) as previously described (48). One of the HCAI outbreak isolates (MBT16-008) failed to grow and therefore could not be analyzed by WGS. Total DNA concentration was estimated using a Qubit fluorometer and a double-stranded DNA (dsDNA) high-sensitivity (HS) assay kit (both ThermoFisher). Fifty nanograms of DNA was prepared using the NEBNext Ultra II FS DNA library prep kit for Illumina (New England Biolabs), and post-PCR cleanup was carried out using AMPure XP beads (Beckman). Library size was validated at approximately 300 bp using the Agilent 2200 TapeStation with an Agilent D1000 ScreenTape system (Willoughby, Australia), and 75-bp paired-end reads were sequenced on a NextSeq 550 system (Illumina).

**FTIR.** The 31 *A. baumannii* isolates were recovered from cryostorage onto prepoured tryptic soy agar (VWR) and incubated aerobically at 37°C for 24 h. A second passage from a single colony of each isolate was performed, and after 24 h, an overloaded 1-$\mu$L loop of confluent growth was collected. The cells were added to a 1.5-mL suspension vial containing inert metal cylinders (Bruker IR Biotyper kit) and 50 $\mu$L 70% (vol/vol) ethanol and vortexed to obtain a homogenous suspension. Fifty microliters of HiPerSov Chromanorm water (VWR) was added, and each vial was vortexed for 1 min. Quintuple 15-$\mu$L spots of each isolate were pipetted onto a 96-spot microtiter plate. Duplicate 12-$\mu$L spots of Bruker infrared test standards 1 and 2 (IRTS1 and IRTS2) were included during each run. The standards were prepared according to the manufacturer's instructions. The microtiter plate was dried above a 37°C hot plate for 30 min, and strain typing was performed using an IR Biotyper and the IR Biotyper software version 2.1.0.195 (Bruker UK).

**Data analysis. (i) Bruker FlexAnalysis method.** Following acquisition each of the 27 spectra (per isolate) was analyzed as described in the MALDI Biotyper protocol using the MBT_standard.FAMSMethod in Bruker FlexAnalysis (version 3.4). Peaks were detected using a centroid algorithm within a 2.0-*m/z* width range. Curve smoothing was performed using a Savitzky-Golay algorithm (44). Replicate spectra were visually inspected, and any peaks below 500 arbitrary units (a.u.) or those deviating outside a mass tolerance of $\pm\sim$0.025% of the estimated *m/z* value were excluded. Mass peak lists for all isolates were recorded in Excel (Microsoft), and peaks potentially representing unique biomarkers for individual strains were recorded. Peaks that satisfied the following criteria were considered strain-specific biomarkers: (i) above 500 a.u. for at least two out of three technical replicate spectra, (ii) at least 5.0-*m/z* (Da) difference from peaks of a similar size, and (iii) present for at least one but not all of the 18 isolates.

**(ii) Bioinformatic analysis of MALDI-TOF MS spectra.** Processed spectral files (referred to as mzXml files) were exported from FlexAnalysis for each isolate for analysis in BioNumerics software (Applied Maths, version 7.6). Peak matching was performed in BioNumerics based on *m/z* data with a constant tolerance of 0.5, linear tolerance of 300 ppm, and a detection rate of 50 new peak classes per spectrum. Similarity matrices were generated based on the Pearson correlation coefficient, and isolates were clustered using the unweighted pair group method with arithmetic mean (UPGMA). The cophenetic correlation between isolates was calculated and expressed as a percentage on the dendrogram. For each isolate, peak classes with intensity values were exported into MS Excel for further analysis.

Confirmatory bioinformatic analysis of spectra was performed using Clover MS data analysis software (Clover BioSoft). Spectra were summarized for each isolate, and a peak matrix was generated. Summarized spectra were clustered by UPGMA.

**(iii) Calculating variability in peak height as a metric for reproducibility of MALDI-TOF MS spectra.** Using the numerical data exported into MS Excel from BioNumerics, two peak classes that were common to all 18 *A. baumannii* isolates (5178 *m/z* and 5751 *m/z*) and one peak that could represent a strain-specific biomarker (3723 *m/z*) were identified. Peak height (intensity) values were normalized to the total intensity for all spectra, and a mean value with relative standard deviation was calculated. This was performed for triplicate technical replicates, triplicate "spots" of cocrystallized matrix and protein extract, and triplicate days of experiments. A mean of the relative standard deviation (RSD) at each of these three time points was calculated for each isolate and plotted.

**(iv) Whole-genome phylogenetic analysis.** Generated FASTQ files containing paired-end sequences for each isolate were screened against all complete *A. baumannii* reference genomes found on the NCBI RefSeq database using Mash (49) to identify the closest-matching reference sequence. The best-matching genome was *A. baumannii* strain BAL062 (accession number NZ_LT594095), and all samples were mapped to this reference using BBMap (50). Single nucleotide variants (SNVs) were called against the reference using Freebayes (51), and variants were taken forward only if (i) read depth was >5, (ii) mapping quality was >30, (iii) base quality was >20, (iv) alternate read frequency was >80%, (v) there

were >2 reads on both strands, and (vi) there were >2 reads with variants present at both the 5′ and 3′ ends of the fragments. Variant positions were also masked if not present at >5-read depth in 90% of samples. Possible recombination sites were identified and masked using Gubbins (52), and a maximum likelihood phylogenetic tree was inferred as part of an established bioinformatic workflow (53) from the aligned variant sites using RAxML under the GTRCAT model (54).

**(v) FTIR analysis.** Data were analyzed using IR Biotyper software version 2.1.0.195 (Bruker UK) according to the manufacturer's recommended workflow.

**Data availability.** MALDI-TOF MS spectra generated for strain typing are available as processed mzXml files at https://platform.clovermsdataanalysis.com/repository/collection/ABAU001. MALDI-TOF MS FlexAnalysis mass peak lists are included in File S3 in the supplemental material. Whole-genome sequencing (WGS) data sets generated and/or analyzed during the current study are available in the European Nucleotide Archive repository, https://www.ebi.ac.uk/ena/browser/view/PRJEB59585. Fourier transform infrared spectroscopy (FTIR) data sets are available at https://platform.clovermsdataanalysis .com/repository/collection/ABAU002.

## SUPPLEMENTAL MATERIAL

Supplemental material is available online only.
**SUPPLEMENTAL FILE 1**, XLSX file, 0.01 MB.
**SUPPLEMENTAL FILE 2**, PNG file, 0.1 MB.
**SUPPLEMENTAL FILE 3**, XLSX file, 0.1 MB.
**SUPPLEMENTAL FILE 4**, PDF file, 0.5 MB.

## ACKNOWLEDGMENTS

This work was supported by the UK National Measurement System and the European Metrology Program for Innovation and Research (EMPIR) joint research project [HLT07] "AntiMicroResist," which has received funding from the EMPIR program cofinanced by the Participating States and the European Union's Horizon 2020 research and innovation program.

Acknowledgment is given to Indran Balakrishnan, Gemma Vanstone, and Kerry Roulston (Royal Free London) for sourcing patient isolates and to Giuseppina Maniscalco for assistance with collecting and culturing the organisms. Acknowledgment is given to the infection control nurses at the Royal Free London NHS Foundation Trust for their assistance with gathering epidemiological and patient data. We are grateful to Vanya Gant, Virve (Vicky) Enne, and the INHALE study group for provision of clinical isolates. We are grateful to members of the Department of Microbiology, Virology and Infection Control, Great Ormond Street Hospital for Children NHS Foundation Trust, for assistance with data acquisition. We are grateful to Jane Turton (PHE) for critical review of the manuscript and to Mumin Eminov and Ditte Find (Bruker) for their assistance with the FTIR analysis. We gratefully acknowledge Luis Mancera, Manuel J. Arroyo, and Jesús Jiménez (Clover Bioanalytical Software) for providing access to software for analysis of MALDI-TOF MS spectra and for visualization of SNV data.

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
