## [Reviewer comments · Microbiology Spectrum]

Microbiology Spectrum

Evaluation of MALDI-TOF MS for molecular typing of *Acinetobacter baumannii* in comparison with orthogonal methods

Eloise Busby, Ronan Doyle, Clara Leboreiro Babe, Kathryn Harris, Damien Mack, Gema Méndez, Denise O'Sullivan, Vicky Pang, Zahra Sadouki, Priya Solanki, Jim Huggett, Timothy McHugh, and Emmanuel Wey

Corresponding Author(s): Emmanuel Wey, Royal Free London NHS Foundation Trust

Review Timeline:

Submission Date:	December 5, 2022
Editorial Decision:	January 3, 2023
Revision Received:	March 23, 2023
Accepted:	March 23, 2023

Editor: Paul Luethy

Reviewer(s): Disclosure of reviewer identity is with reference to reviewer comments included in decision letter(s). The following individuals involved in review of your submission have agreed to reveal their identity: Aliaa Gamaleldin Aboulela (Reviewer #1); Xavier Alexander (Reviewer #2)

Transaction Report:

DOI: <https://doi.org/10.1128/spectrum.04995-22>

January 3, 2023

Dr. Emmanuel Wey
Royal Free London NHS Foundation Trust
London
United Kingdom

Re: Spectrum04995-22 (Evaluation of MALDI-TOF MS for molecular typing of *Acinetobacter baumannii* in comparison with orthogonal methods)

Dear Dr. Emmanuel Wey:

Link Not Available

Sincerely,

Paul Luethy

Journals Department
Reviewer comments:

Reviewer #1 (Comments for the Author):

In this article, the authors evaluated two approaches for analyzing the data from MALDI-TOF MS, to evaluate its potential application as a rapid bacterial typing tool that would offer timely data about the clonal relatedness of outbreak isolates. To this end, they compared the typing output from MALDI-TOF MS (the test method), with that obtained from WGS and FTIR. The abstract included a concluding statement "Emerging typing methods which offer different degrees of resolution and target different biomolecules, could be utilized in-house to more quickly confirm or rule out transmission events." This statement is redundant and ill-defined, as it does not specify the exact methods by name. It is recommended to be more specific while referring to applicable alternatives. I recommend rephrasing this statement to deliver the intended meaning more clearly.

The authors included a group of 13 additional random isolates as a control group. However, they did not test this group by the test method under study; MALDI-TOF MS. Instead, they tested these isolates with the two comparative methods. I think testing all isolates by MALDI-TOF MS would add more value to the results, especially since the final conclusion about MALDI-TOF MS typing was its being unsatisfactory when used alone with the default software for MALDI-TOF MS analysis; FlexAnalysis. Testing the control strains by the same 2 approaches of data analysis would provide clearer visualization of how far the MALDI-TOF MS typing can be reliable/unreliable in discriminating totally unrelated strains.

The authors used "BioNumerics" and "Clover MS data analysis software" for the analysis of MALDI-TOF MS output data and the UPGMA clustering output of both software tools was almost the same, even though this information needed careful text searching and looking at both Figure 2a and Figure S1 to get it. I think it would be good enough to mention in the legend of figure 2 that the UPGMA represented is the output of BioNumerics software, and to refer to Figure S1 for checking the output of Clover MS clustering. Also, the authors need to give a final resume about their user experience, the cost, and the added value of using "Clover MS data analysis software", in relation to "BioNumerics", as this would help future researchers select one of them to pursue the work with and further validate or decline the approach of using bioinformatics for bacterial typing by MALDI-TOF MS, since the reproducibility of the method is highly affected by the data analysis approach.

Regarding the phylogenetic analysis of WGS data by the maximum likelihood approach, what is the justification for that? and why did not the authors use the same distance-based approach; UPGMA, instead? In my opinion, this would better reveal the relatedness between the isolates, and would be more comparable with the dendrogram outputs from "BioNumerics" and "Clover Biosoft". Also, Figure 2b shows the results of only 17 isolates out of a total of 31 whole genome- sequenced isolates including test cases and controls. I think including non-outbreak isolates in the dendrogram would be more informative. Nevertheless, since the data of reference genomes were already used for the alignment, it would be good to show the reference genome in the dendrogram as well, particularly if the maximum likelihood approach is chosen, as this would elaborate the evolutionary relationship between the circulating isolates and the reference strain genome.

As for FTIR, the authors provided results in a discrete and simple approach. However, for better comparability with the other 2 approaches, it would be useful to analyze FTIR output data with an additional similar bioinformatics approach, with the inclusion of data from outbreak and non-outbreak isolates.

Regarding figure S2a which displays the results of the reference lab, I think data can be represented in a clearer way other than the arrows. For instance, the VNTR profile may be simply represented alongside the PFGE-type, with color coding of related isolates.

I recommend calculation of the concordance between the non-phylogenetic outputs (Biotypes) of the typing methods MALDI-TOF MS and FTIR with the typing results of PFGE and VNTR using the online tool: "Comparing partitions website"; <http://www.comparingpartitions.info/index.php?link=Tut2>.

Finally, it would be useful that the authors state clearly all the possible sources of variability they encountered with MALDI-TOF MS typing, for which they found it unreliable as a sole method for rapid in-house strain typing while suggesting clear steps for standardization of the testing procedure to minimize variation to the least possible, so that the method may be further validated with the implementation of either of "BioNumerics" or "Clover Biosoft" bioinformatics approach, whichever they recommend to be more preferable and user friendly.

Reviewer #2 (Comments for the Author):

Experiments were designed to describe the hypothesis and the results were satisfactory.

Reviewer #3 (Comments for the Author):

The authors would like to evaluate the utility of MALDI-TOF for molecular typing of *Acinetobacter baumannii* and compare the results with WGS and FTIR. However, only the outbreak isolates were characterised by the 3 methods but not the control isolates. The classification of the isolates as related or not related to the outbreak is ambiguous and the presence of isolates belongs to the same patient further complicate the analysis. All these should be presented and discussed as a whole in order to derive a conclusive finding. A larger sample size with a more thorough and definite analysis is required.

1. Line 105-107, please indicate which isolates are from the same patients?
2. Line 115, the authors mentioned that 13 isolates were unrelated to the outbreak but some of these isolates (Abau-Iso2-11/19/26/23/20/27) seems to have clustered with the outbreak strains which indicated they are linked to the outbreak. How the authors rule out this possibility?
3. Line 119, please indicate which 3 isolates?
4. Line 121-125, the AST results should be presented along with the resistance genes found in WGS.
5. Line 144, is there any recommendation that the samples can be kept for days and remeasured by the manufacturer? How the samples are kept?
6. WGS and FTIR have been performed on all the 31 isolates but PFGE, VNTR and MALDI-TOF have only been performed on 18 isolates. All the isolates should have been examined thoroughly by all the methods for comparison.
7. The assembled genome sequences of the isolates should be deposited into GenBank with the accession numbers provided in the manuscript.

8. Figure 4, the spectrum name is missing.

Staff Comments:

Preparing Revision Guidelines

Please return the manuscript within 60 days; if you cannot complete the modification within this time period, please contact me. If you do not wish to modify the manuscript and prefer to submit it to another journal, please notify me of your decision immediately so that the manuscript may be formally withdrawn from consideration by Microbiology Spectrum.

Reviewer's report

“Evaluation of MALDI-TOF MS for molecular typing of *Acinetobacter baumannii* in comparison with orthogonal methods”

In this article, the authors evaluated two approaches for analyzing the data from MALDI-TOF MS, to evaluate its potential application as a rapid bacterial typing tool that would offer timely data about the clonal relatedness of outbreak isolates. To this end, they compared the typing output from MALDI-TOF MS (the test method), with that obtained from WGS and FTIR.

The abstract included a concluding statement **“Emerging typing methods which offer different degrees of resolution and target different biomolecules, could be utilized in-house to more quickly confirm or rule out transmission events.”**

This statement is redundant and ill-defined, as it does not specify the exact methods by name. It is recommended to be more specific while referring to applicable alternatives. I recommend rephrasing this statement to deliver the intended meaning more clearly.

The authors included a group of 13 additional random isolates as a control group. However, they did not test this group by the test method under study; MALDI-TOF MS. Instead, they tested these isolates with the two comparative methods. I think testing all isolates by MALDI-TOF MS would add more value to the results, especially since the final conclusion about MALDI-TOF MS typing was its being unsatisfactory when used alone with the default software for MALDI-TOF MS analysis; FlexAnalysis. Testing the control strains by the same 2 approaches of data analysis would provide clearer visualization of how far the MALDI-TOF MS typing can be reliable/unreliable in discriminating totally unrelated strains.

The authors used “BioNumerics” and “Clover MS data analysis software” for the analysis of MALDI-TOF MS output data and the UPGMA clustering output of both software tools was almost the same, even though this information needed careful text searching and looking at both Figure 2a and Figure S1 to get it. I think it would be good enough to mention in the legend of figure 2 that the UPGMA represented is the output of BioNumerics software, and to refer to Figure S1 for checking the output of Clover MS clustering. Also, the authors need to give a final resume about their user experience, the cost, and the added value of using “Clover MS data analysis software”, in relation to “BioNumerics”, as this would help future researchers select one of them to pursue the work with and further validate or decline the approach of using bioinformatics for bacterial typing by MALDI-TOF MS, since the reproducibility of the method is highly affected by the data analysis approach.

Regarding the phylogenetic analysis of WGS data by the maximum likelihood approach, what is the justification for that? and why did not the authors use the same distance-based approach; UPGMA, instead? In my opinion, this would better reveal the relatedness between the isolates, and would be more comparable with the dendrogram outputs from “BioNumerics” and “Clover Biosoft”. Also, Figure 2b shows the results of only 17 isolates out of a total of 31 whole genome-sequenced isolates including test cases and controls. I think including non-outbreak isolates in the dendrogram would be more informative. Nevertheless, since the data of reference genomes were already used for the alignment, it would be good to show the reference genome in the dendrogram as well, particularly if the maximum likelihood approach is chosen, as this would elaborate the evolutionary relationship between the circulating isolates and the reference strain genome.

As for FTIR, the authors provided results in a discrete and simple approach. However, for better comparability with the other 2 approaches, it would be useful to analyze FTIR output data with an additional similar bioinformatics approach, with the inclusion of data from outbreak and non-outbreak isolates.

Regarding figure S2a which displays the results of the reference lab, I think data can be represented in a clearer way other than the arrows. For instance, the VNTR profile may be simply represented alongside the PFGE-type, with color coding of related isolates.

I recommend calculation of the concordance between the non-phylogenetic outputs (Biotypes) of the typing methods MALDI-TOF MS and FTIR with the typing results of PFGE and VNTR using the online tool: “Comparing partitions website”;
<http://www.comparingpartitions.info/index.php?link=Tut2>.

Finally, it would be useful that the authors state clearly all the possible sources of variability they encountered with MALDI-TOF MS typing, for which they found it unreliable as a sole method for rapid in-house strain typing while suggesting clear steps for standardization of the testing procedure to minimize variation to the least possible, so that the method may be further validated with the implementation of either of “BioNumerics” or “Clover Biosoft” bioinformatics approach, whichever they recommend to be more preferable and user friendly.

Dear Editorial review team,

Thank you for your review of our original submitted manuscript on the Evaluation of MALDI-TOF MS for molecular typing of *Acinetobacter baumannii* in comparison with orthogonal methods.

We welcome the constructive and insightful comments on the first submitted version of the manuscript. We have listed below our responses [R] and amendments specific to the reviewer's comments.

Reviewer #1 (Comments for the Author):

In this article, the authors evaluated two approaches for analyzing the data from MALDI-TOF MS, to evaluate its potential application as a rapid bacterial typing tool that would offer timely data about the clonal relatedness of outbreak isolates. To this end, they compared the typing output from MALDI-TOF MS (the test method), with that obtained from WGS and FTIR.

The abstract included a concluding statement "Emerging typing methods which offer different degrees of resolution and target different biomolecules, could be utilized in-house to more quickly confirm or rule out transmission events."

This statement is redundant and ill-defined, as it does not specify the exact methods by name. It is recommended to be more specific while referring to applicable alternatives. I recommend rephrasing this statement to deliver the intended meaning more clearly.

[R] We have amended this statement in the abstract.

The authors included a group of 13 additional random isolates as a control group. However, they did not test this group by the test method under study; MALDI-TOF MS. Instead, they tested these isolates with the two comparative methods. I think testing all isolates by MALDI-TOF MS would add more value to the results, especially since the final conclusion about MALDI-TOF MS typing was its being unsatisfactory when used alone with the default software for MALDI-TOF MS analysis; FlexAnalysis. Testing the "control" strains by the same 2 approaches of data analysis would provide clearer visualization of how far the MALDI-TOF MS typing can be reliable/unreliable in discriminating totally unrelated strains.

[R] We have removed 'control' (swapped in 'reference') and added additional wording in 'Materials and methods: Drug resistant clinical isolates' to further describe these 13 isolates. The time span of the collection of the additional isolates is also included in text (2011 to 2017) and in Suppl table S1. We have referred to the INHALE study in the text (Ref. 34) as the source of 3 of the additional 13 isolates. These reference strains represent temporal and geographically relevant isolates from clinical cohorts isolated from the same institution which were available to root the outbreak strains. When we use the term 'outbreak' we are using the term under its United

Kingdom Health Security Agency (UKHSA) (previously known as PHE) definition of a healthcare associated infection outbreak defined as: "Two or more linked cases with the same infectious agent associated with the same healthcare setting over a specified time period or A higher than expected number of cases of HAI in a given healthcare area over a specified time period." Due to resource restrictions on the availability of the MALDI-TOF MS platform, permissions required from the orthogonal INHALE study, and restrictions on resource availability on VNTR and PFGE typing by UKHSA, it was not possible for the additional 13 reference strains to undergo PFGE, VNTR and MALDI-TOF MS.

The authors used "BioNumerics" and "Clover MS data analysis software" for the analysis of MALDI-TOF MS output data and the UPGMA clustering output of both software tools was almost the same, even though this information needed careful text searching and looking at both Figure 2a and Figure S1 to get it. I think it would be good enough to mention in the legend of figure 2 that the UPGMA represented is the output of BioNumerics software, and to refer to Figure S1 for checking the output of Clover MS clustering.

[R] We have referred to Figure S1 in the Figure 2 legend.

Also, the authors need to give a final resume about their user experience, the cost, and the added value of using "Clover MS data analysis software", in relation to "BioNumerics", as this would help future researchers select one of them to pursue the work with and further validate or decline the approach of using bioinformatics for bacterial typing by MALDI-TOF MS, since the reproducibility of the method is highly affected by the data analysis approach.

[R] Gema Mendez is an author and employed by Clover Biosoft. Emmanuel Wey, the last author, is on the advisory board of Clover Biosoft and although he has not and does not receive any remuneration it could be perceived as a potential conflict of interest to provide a favorable comparison in relation to the use of the Clover Biosoft platform. We have, at the reviewer's request, added that both platforms were of equivalent cost and subjective ease of use in Lines 391-393 of the manuscript.

Regarding the phylogenetic analysis of WGS data by the maximum likelihood approach, what is the justification for that? and why did not the authors use the same distance-based approach; UPGMA, instead? In my opinion, this would better reveal the relatedness between the isolates, and would be more comparable with the dendrogram outputs from "BioNumerics" and "Clover Biosoft".

[R] The maximum likelihood approach is part of Dr Ronan Doyle's established pipeline used at the time which is now cited in the text (Ref. 43). In addition, the WGS files are now publicly available on the ENA website (link given in text). In addition, the MALDI-TOF MS and FTIR files are now publicly available on the clover data analysis platform (link given in text), lines 244-252 (Section Data availability).

Also, Figure 2b shows the results of only 17 isolates out of a total of 31 whole genome- sequenced isolates including test cases and controls. I think including non-outbreak isolates in the dendrogram would be more informative.

[R] Figure 2b presents a comparison between the different methods used to type the HCAI isolates (MALDI, WGS, FTIR), whereas Figure 3 presents a WGS comparison between the HCAI and 13 additional non-outbreak isolates. Text amended to explain that 17 out of a possible 18 HCAI isolates were analysed by WGS because one isolate (MBT16-008) failed to grow (line 168-169). In addition, WGS, MALDI-TOF MS and FTIR data files are now publicly available for use (links given in text; Data availability).

Nevertheless, since the data of reference genomes were already used for the alignment, it would be good to show the reference genome in the dendrogram as well, particularly if the maximum likelihood approach is chosen, as this would elaborate the evolutionary relationship between the circulating isolates and the reference strain genome.

[R] The WGS analysis focused specifically on the comparison between the HCAI and non-outbreak samples. The same reference genome was not analysed by MALDI or FTIR, so inclusion in the WGS phylogenetic tree could be of limited value for the comparison. In addition, the reference genome is not part of the original outbreak and probably separated from the other samples by years of evolution and would likely cluster separately. WGS data files are now available publicly (link in text; Data availability).

As for FTIR, the authors provided results in a discrete and simple approach. However, for better comparability with the other 2 approaches, it would be useful to analyze FTIR output data with an additional similar bioinformatics approach, with the inclusion of data from outbreak and non-outbreak isolates.

[R] Following advice from the manufacturer (Bruker), a standard method for FTIR data analysis was implemented for the study. This has now been stated in the manuscript (line 242-243).

Regarding figure S2a which displays the results of the reference lab, I think data can be represented in a clearer way other than the arrows. For instance, the VNTR profile may be simply represented alongside the PFGE-type, with color coding of related isolates.

[R] S2a has been removed and the relevant data added to Table S1.

I recommend calculation of the concordance between the non-phylogenetic outputs (Biotypes) of the typing methods MALDI-TOF MS and FTIR with the typing results of PFGE and VNTR using the online tool: "Comparing partitions website"; <http://www.comparingpartitions.info/index.php?link=Tut2>.

[R] We have addressed this suggestion by removing the rather convoluted Figure S2, and also by listing all the biotypes alongside patient data in table S1. We have also added supplemental files S1 and S2 containing numerical SNV comparative tables and heat maps for the isolates. All 18 isolates were classified as belonging to International Clone II lineage OXA-23 clone 1. PFGE data and VNTR profiles are included in Table S1.

Finally, it would be useful that the authors state clearly all the possible sources of variability they encountered with MALDI-TOF MS typing, for which they found it unreliable as a sole method for rapid in-house strain typing while suggesting clear steps for standardization of the testing procedure to minimize variation to the least possible, so that the method may be further validated with the implementation of either of "BioNumerics" or "Clover Biosoft" bioinformatics approach, whichever they recommend to be more preferable and user friendly.

[R] A discussion around the specific issues with the MALDI-TOF MS typing workflow is included in the text, which predominantly relate to the subjectivity of the FlexAnalysis method, and the hypothesis that variability exists in the co-crystallization of matrix and proteins (Ref. 45) between different replicates and days (line 343-348, line 369-372, line 377-378, line 383-386). Gema Mendez is an author and employed by Clover Biosoft, and Emmanuel Wey is on the advisory board of Clover Biosoft. Although he has not and does not receive any remuneration it could be perceived as a potential conflict of interest to provide a favorable comparison in relation to the use of the CloverBiosoft platform. We have, at the reviewer's request, added that both platforms were of equivalent cost and subjective ease of use and each had its strengths and weaknesses Lines 391-393.

Reviewer #2 (Comments for the Author):

Experiments were designed to describe the hypothesis and the results were satisfactory.

[R] We thank the reviewer for their comments.

Reviewer #3 (Comments for the Author):

The authors would like to evaluate the utility of MALDI-TOF for molecular typing of *Acinetobacter baumannii* and compare the results with WGS and FTIR. However, only the outbreak isolates were characterised by the 3 methods but not the control isolates. The classification of the isolates as related or not related to the outbreak is ambiguous and the presence of isolates belongs to the same patient further

complicate the analysis. All these should be presented and discussed as a whole in order to derive a conclusive finding. A larger sample size with a more thorough and definite analysis is required.

[R] Additional text included in Materials and Methods (Line 132-139) to describe the origin of the additional 13 isolates. Time span of additional isolates also included in text (2011 to 2017) and in Suppl table S1. Reference to INHALE study included in text (Ref. 34). The reference strains represent temporal and geographical relevant isolates from clinical cohorts isolated from the same institution which were available to root the outbreak strains. When we use the term 'outbreak' we are using the term under its United Kingdom Health Security Agency UKHSA definition of a healthcare associated infection outbreak defined as: "Two or more linked cases with the same infectious agent associated with the same healthcare setting over a specified time period or A higher than expected number of cases of HAI in a given healthcare area over a specified time period". We have also clarified the relationship of isolates to patients in Table S1.

1. Line 105-107, please indicate which isolates are from the same patients?

[R] Isolates MBT16-005 and MBT16-039 were obtained from the same patient (Patient ID: 2), and MBT16-025 and MBT16-031 were obtained from Patient ID: 3. This information is now included in Supplementary Table S1.

2. Line 115, the authors mentioned that 13 isolates were unrelated to the outbreak but some of these isolates (Abau-Iso2-11/19/26/23/20/27) seems to have clustered with the outbreak strains which indicated they are linked to the outbreak. How the authors rule out this possibility?

[R] We have included a line referring to (and citation of) UKHSA clinical definition of outbreak in the text (Ref. 32). Description of temporal origin of isolates listed in Table S1, and citation of The INHALE study included in the manuscript text (Ref. 34).

3. Line 119, please indicate which 3 isolates?

[R] These isolates are described in line 138-139, along with citation of the INHALE study from which they were derived (Ref. 34).

4. Line 121-125, the AST results should be presented along with the resistance genes found in WGS.

[R] We have clarified the text in the manuscript and included 'as part of routine microbiology diagnostic service'. WGS sequences are available in the ENA database (link in manuscript; Data

availability) if resistance genes are sought. MDR was focus of outbreak but not the focus of our study.

5. Line 144, is there any recommendation that the samples can be kept for days and remeasured by the manufacturer? How the samples are kept?

[R] We have clarified in text that the isolates were recovered on fresh agar before each typing experiment.

6. WGS and FTIR have been performed on all the 31 isolates but PFGE, VNTR and MALDI-TOF have only been performed on 18 isolates. All the isolates should have been examined thoroughly by all the methods for comparison.

[R] PFGE and VNTR typing at the time of this work were performed by the UKHSA national reference laboratory on isolates suspected of belonging to an outbreak as defined by UKHSA guidelines. As these isolates did not meet criteria at the time they were not processed, as this was a restricted resource service provided by the National Reference centre. We have addressed the query regarding MALDI-TOF MS analysis in our response to reviewer 1. Due to resource restrictions on the availability of the MALDI-TOF MS platform, permissions required from the orthogonal INHALE study, and restrictions on availability of typing by UKHSA, it was not possible for the additional 13 reference strains to undergo PFGE, VNTR and MALDI-TOF MS.

7. The assembled genome sequences of the isolates should be deposited into GenBank with the accession numbers provided in the manuscript.

[R] Sequences are now available in the European Nucleotide Archive repository, <https://www.ebi.ac.uk/ena/browser/view/PRJEB59585> and the link is provided in the manuscript along with public database locations for the MALDI-TOF MS and FTIR data files.

8. Figure 4, the spectrum name is missing.

[R] Figure 4 has been modified. Spectrum name and cluster purity have been removed as we felt that they did not add anything to the figure.

Kind regards,

Dr Emmanuel Wey

March 23, 2023

Dr. Emmanuel Quintela Wey
Royal Free London NHS Foundation Trust
Department of Infection
Pond Street
Hampstead
London, London NW3 2QG
United Kingdom

Re: Spectrum04995-22R1 (Evaluation of MALDI-TOF MS for molecular typing of *Acinetobacter baumannii* in comparison with orthogonal methods)

Dear Dr. Emmanuel Quintela Wey:

Your manuscript has been accepted, and I am forwarding it to the ASM Journals Department for publication. You will be notified when your proofs are ready to be viewed.

Sincerely,

Paul Luethy
Editor, Microbiology Spectrum
